# Development and evaluation of an artificial intelligence system for COVID-19 diagnosis

Cheng Jin[1,4], Weixiang Chen[1,4], Yukun Cao [2,3,4], Zhanwei Xu[1], Zimeng Tan[1], Xin Zhang[2,3], Lei Deng[1], Chuansheng Zheng [2,3], Jie Zhou[1], Heshui Shi [2,3✉] & Jianjiang Feng [1✉]

Early detection of COVID-19 based on chest CT enables timely treatment of patients and helps control the spread of the disease. We proposed an artificial intelligence (AI) system for rapid COVID-19 detection and performed extensive statistical analysis of CTs of COVID-19 based on the AI system. We developed and evaluated our system on a large dataset with more than 10 thousand CT volumes from COVID-19, influenza-A/B, non-viral community acquired pneumonia (CAP) and non-pneumonia subjects. In such a difficult multi-class diagnosis task, our deep convolutional neural network-based system is able to achieve an area under the receiver operating characteristic curve (AUC) of 97.81% for multi-way classification on test cohort of 3,199 scans, AUC of 92.99% and 93.25% on two publicly available datasets, CC-CCII and MosMedData respectively. In a reader study involving five radiologists, the AI system outperforms all of radiologists in more challenging tasks at a speed of two orders of magnitude above them. Diagnosis performance of chest x-ray (CXR) is compared to that of CT. Detailed interpretation of deep network is also performed to relate system outputs with CT presentations. The code is available at https://github.com/ChenWWWeixiang/diagnosis_covid19.

[1] Department of Automation, Beijing National Research Center for Information Science and Technology, Tsinghua University, Beijing, China. [2] Department of Radiology, Union Hospital, Tongji Medical College, Huazhong University of Science and Technology, Wuhan, China. [3] Hubei Province Key Laboratory of Molecular Imaging, Wuhan, China. [4] These authors contributed equally: Cheng Jin, Weixiang Chen, Yukun Cao. ✉email: heshuishi@hust.edu.cn; jfeng@tsinghua.edu.cn

The SARS-CoV-2 (ref. [1]) has infected >16 million people worldwide, killed >644 thousands (as of the time this article was written) and is still spreading rapidly worldwide. It is important to detect COVID-19 as quickly and accurately as possible for controlling the spread of the disease and treating patients. Even though reverse transcription-polymerase chain reaction (RT-PCR) is still ground truth of COVID-19 diagnosis, the sensitivity of RT-PCR is not high enough for low viral load present in test specimens or laboratory error[2]. In addition, the supply of kits of RT-PCR varies from place to place and some developing countries are in short supply of it[3].

As a result, some countries used chest imaging, such as chest CT or chest x-ray (CXR) as a first-line investigation and patient management tools[4,5]. Chest imaging, especially CT, can show early lesions in the lung and, if diagnosed by experienced radiologists, can achieve high sensitivity. In addition, chest imaging technologies, especially CXR, are widely available and economic. At present, the diagnosis of chest CT depends on visual diagnosis of radiologists, which has some problems. Firstly, chest CT contains hundreds of slices, which takes a long time to diagnose. Secondly, COVID-19, as a new lung disease, has similar manifestations with various types of pneumonia[6]. Radiologists need to accumulate a lot of CT diagnostic experience to achieve a high diagnostic performance, especially in differentiating similar deceases. COVID-19 can still be a threat for a long time, since the situations in some countries are not optimistic. If COVID-19 and influenza were to break out together, which is possible, CT diagnosis workload would likely be far beyond the number of qualified radiologists.

Artificial intelligence (AI) may be the unique preparation to take up this challenge. Powered by large labeled datasets[7] and modern GPUs, AI, especially deep learning technique[8], has achieved excellent performance in several computer vision tasks, such as image classification[9] and object detection[10]. Recent research shows that AI algorithms can even achieve or exceed the performance of human experts in certain medical image diagnosis tasks, including lung diseases[11–17]. Comparing to other lung diseases, such as lung nodule detection[18–20], tuberculosis diagnosis[16,21], and lung cancer screening[15], differentiating COVID-19 from other pneumonias has unique difficulty, i.e., high similarity of pneumonias of different types (especially in early stage) and large variations in different stages of the same type. Hence, developing AI diagnosis algorithm specific to COVID-19 is necessary. The AI diagnosis algorithm also has the advantages of high efficiency, high repeatability, and easy large-scale deployment.

There are already some published studies on CT-based COVID-19 diagnosis systems[22,23]. Here, we briefly review several representative studies employing relatively large datasets. Zhang et al.[24] developed COVID-19 diagnosis system on a database consisting 4154 patients, and it can differentiate COVID-19 from other common pneumonias and normal healthy with AUC of 0.9797. In their system, the classification was based on lesion segmentation result, and the lesion segmentation DICE index was ~0.662, which is not an accurate representation of lesions. Another drawback is manual annotation of training segmentation masks is a very expensive procedure. Li et al.[25] developed an AI system and yielded AUC of 0.96 for COVID-19 detection on dataset consisting of 3322 subjects, including COVID-19, CAP, and healthy people. Their system extracted features on slices and fused them into volume level, which increased much memory demand, while without extracting more informative 3D features. Several slice-level diagnosis methods[17,26,27] were proposed which were quite similar to Li et al.'s work. Some AI systems employed 3D convolution neural networks, but only considered the relatively simple two-category classification[28,29]. There are also a few COVID-19 detection

systems using CXR[30], but the number of subjects with COVID-19 in these studies is much smaller than that in the studies using CT, and no study has quantitively compared performances of CXR and CT using paired data.

In this work, we construct a clinically representative large-scale dataset with 11,356 CT scans from three centers in China and four publicly available databases, which is much larger than previous studies. To understand relative performances of CT and CXR for detecting COVID-19, we develop both CT-based and CXR-based diagnosis systems, and test them using paired data, which has not been studied before. We compare the diagnostic performance of our CT-based diagnosis system with that of five radiologists in reader studies, and the results show that the performance of this system is higher than that of experienced radiologists. In addition, based on prediction score on every slice of CT volume, we locate the lesion areas in COVID-19 patients and perform a statistical study of different subsets of patients. The specific phenotypic basis of the diagnosis output is also traced by an interpretation network, and radiomics analysis is applied to understand the imaging characteristics of COVID-19.

## Results

**Datasets for system development and evaluation**. We developed and evaluated a deep learning-based COVID-19 diagnosis system, using multi-class multicenter data, which included 11,356 CT scans from 9025 subjects consisting of COVID-19, CAP, influenza, and non-pneumonia. CAP subjects included in our database were all nonviral CAP. Data were collected in three different centers in Wuhan, and from four publicly available databases, LIDC–IDRI[31], Tianchi-Alibaba[32], MosMedData[33], and CC-CCII[24] (described in Table 1 and "Methods").

As multistage CT scans of the same person might be similar, the cohort division was performed on subjects with no overlapping subjects in different sub-cohorts. Except MosMedData and CC-CCII, all remaining data were divided into two independent parts, a training cohort of 2688 subjects and a test cohort of 2688 subjects. Three reader study cohorts were randomly chosen from test cohort with respectively 100, 100, and 50 subjects for three tasks, differentiating pneumonia from healthy, differentiating COVID-19 from CAP, and differentiating COVID-19 from influenza. CC-CCII and MosMedData database were used as independent test cohorts containing respectively 2539 subjects and 1110 subjects (described in "Methods").

**Construction of the AI system for COVID-19 diagnosis**. We proposed a deep learning-based AI system for COVID-19 diagnosis, which directly takes CT data as input, to perform lung segmentation, COVID-19 diagnosis, and COVID-infectious slices locating. In addition, we hope that the diagnosis results of AI system can be quantitatively explained in the original image to alleviate the drawback of deep neural networks as a black box. The system consists of five key parts (Fig. 1a), (1) lung segmentation network, (2) slice diagnosis network, (3) COVID-infectious slice locating network, (4) visualization module for interpreting the attentional region of deep networks, and (5) image phenotype analysis module for explaining the features of the attentional region.

The workflow of deep learning-based diagnosis model is shown in Fig. 1b. CT volumes were divided into different cohorts. Then after slice-level training, our model can accurately classify input slices into four categories, including non-pneumonia, CAP, influenza-A or B, and COVID-19. Subsequently, a task-specific fusion module was utilized to fuse slice results into case-level diagnosis according to different diagnosis tasks such that the network can be used in different tasks without retraining. The

**Table 1 Characteristics of patients from our datasets.**

**a**

| # Subject (# Scan) | Non-pneumonia | CAP | Influenza-A/B | COVID-19 | In total |
|---|---|---|---|---|---|
| Training cohort | 1230 (1233) | 666 (666) | 41 (70) | 751 (1294) | 2688 (3263) |
| Test cohort | 1229 (1234) | 666 (668) | 42 (62) | 751 (1235) | 2688 (3199) |
| CC-CCII test cohort | 829 (1079) | 964 (1528) | 0 | 726 (1257) | 2539 (3784) |
| MosMedData test cohort | 254 (254) | 0 | 0 | 856 (856) | 1110 (1110) |
| In total | 3562 (3818) | 2296 (2862) | 83 (132) | 3084 (4542) | 9025 (11356) |

**b**

| # Subject | Non-pneumonia | | CAP | | Influenza-A/B | | COVID-19 | |
|---|---|---|---|---|---|---|---|---|
| | Training cohort | Test cohort | Training cohort | Test cohort | Training cohort | Test cohort | Training cohort | Test cohort |
| Gender | | | | | | | | |
| Male | 25 | 34 | 404 | 424 | 27 | 27 | 348 | 369 |
| Female | 104 | 95 | 262 | 244 | 14 | 15 | 403 | 382 |
| Age | | | | | | | | |
| ≤20 | 0 | 1 | 62 | 69 | 2 | 0 | 1 | 3 |
| 21–40 | 122 | 118 | 116 | 98 | 5 | 7 | 139 | 116 |
| 41–60 | 6 | 10 | 193 | 198 | 7 | 8 | 268 | 261 |
| 61–80 | 1 | 0 | 241 | 243 | 19 | 20 | 314 | 335 |
| >80 | 0 | 0 | 45 | 50 | 8 | 7 | 29 | 36 |
| No record | 0 | 0 | 9 | 10 | 0 | 0 | 0 | 0 |
| # Stage | | | | | | | | |
| 1 | 129 | 129 | 666 | 664 | 28 | 27 | 323 | 341 |
| 2 | 0 | 0 | 0 | 4 | 7 | 11 | 300 | 301 |
| 3 | 0 | 0 | 0 | 0 | 1 | 1 | 103 | 87 |
| >3 | 0 | 0 | 0 | 0 | 5 | 3 | 25 | 22 |

a Number of subjects and scans of each class. b Detailed characteristics of subjects of training and test cohort from Wuhan. Note: as clinical characteristics of publicly available databases are not available due to anonymization, Table b lists only the data from Wuhan.

**Fig. 1 Workflows of the whole study and AI system. a** Workflow of the whole study. **b** Construction and usage of the AI system.

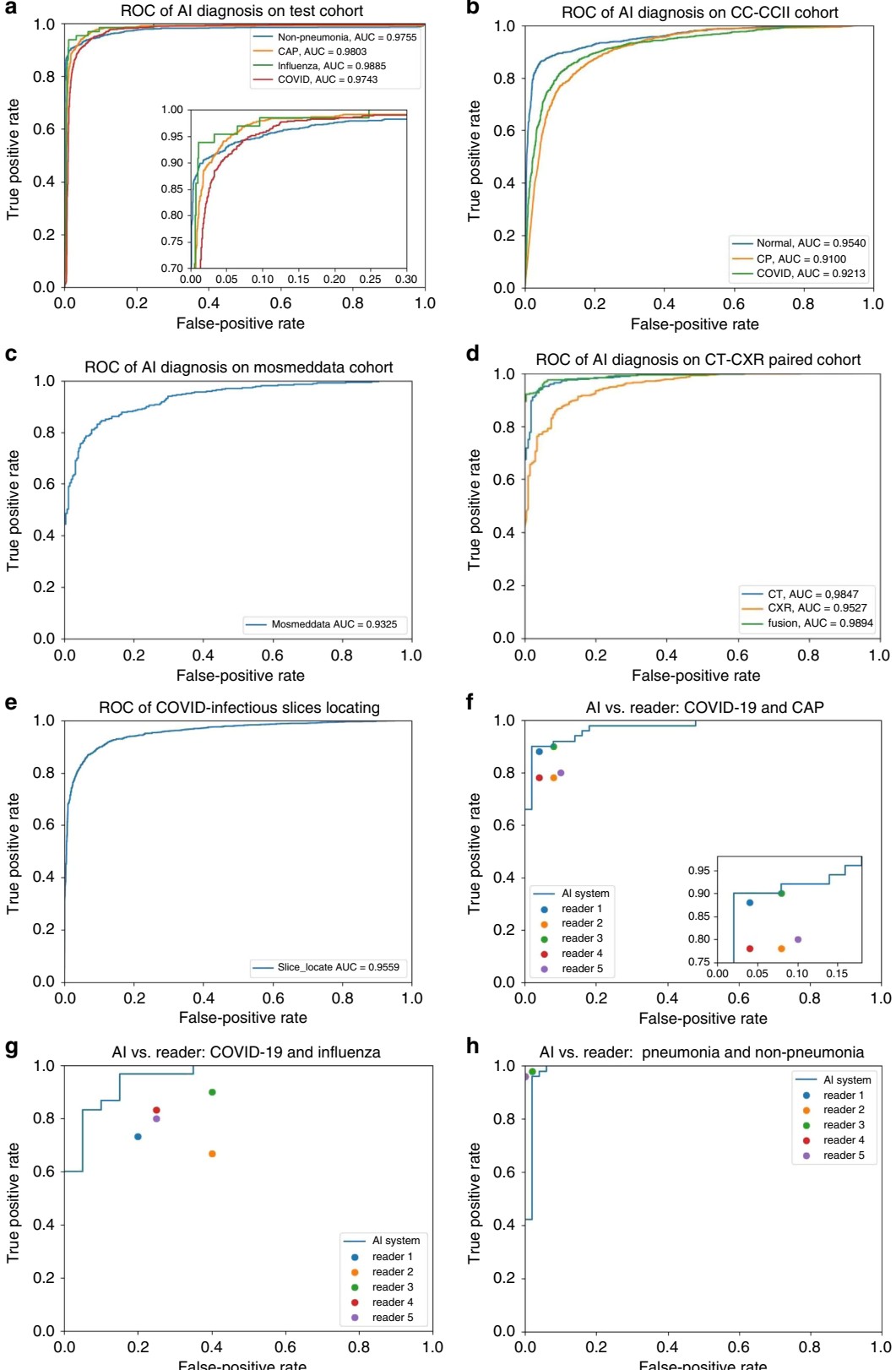

**Fig. 2 Receiver operating curves of the AI system. a** ROC curves of AI system on test cohort. **b** ROC curves of AI system on CC-CCII test cohort. **c** ROC curve of AI system on MosMedData test cohort. **d** ROC curves of CT-based AI system and CXR-based AI system on sub-cohort of test cohort, which has paired CT and CXR data. **e** ROC curve of AI system on COVID-infectious locating. **f** ROC curve together with reader performances on CAP-or-COVID cohort. **g** ROC curve together with reader performances on influenza-or-COVID cohort. **h** ROC curve together with reader performances on pneumonia-or-non-pneumonia cohort.

**Table 2 Metrics of the AI system.**

| Task | AUC (95% CI) | Sensitivity (95% CI) | Specificity (95% CI) |
|---|---|---|---|
| **a** | | | |
| Non-pneumonia diagnosis | 0.9752 (0.9726–0.9783) | 0.9343 (0.9290–0.9429) | 0.9801 (0.9778–0.9827) |
| CAP diagnosis | 0.9804 (0.9776–0.9837) | 0.9687 (0.9634–0.9741) | 0.9407 (0.9366–0.9448) |
| Influenza-A/B diagnosis | 0.9885 (0.9861–0.9928) | 0.8307 (0.7962–0.8696) | 0.9945 (0.9936–0.9960) |
| COVID-19 diagnosis | 0.9745 (0.9722–0.9771) | 0.8703 (0.8620–0.8784) | 0.9660 (0.9629–0.9693) |
| Multi-way metrics | 0.9781 (0.9756–0.9804) | 0.9151 (0.9115–0.9193) (the metric changed to accuracy) | |
| **b** | | | |
| Normal diagnosis | 0.9541 (0.9511–0.9574) | 0.8561 (0.8471–0.8661) | 0.9524 (0.9494–0.9563) |
| Common pneumonia diagnosis | 0.9098 (0.9058–0.9139) | 0.8823 (0.8759–0.8904) | 0.8685 (0.8628–0.8745) |
| COVID-19 diagnosis | 0.9212 (0.9175–0.9255) | 0.7799 (0.7706–0.7893) | 0.9355 (0.9315–0.9397) |
| Multi-way metrics | 0.9299 (0.927–0.933) | 0.8435 (0.8391–0.8483) (the metric changed to accuracy) | |
| **c** | | | |
| COVID-19 diagnosis on MosMedData cohort | 0.9325 (0.9257–0.9382) | 0.9446 (0.9379–0.9510) | 0.6613 (0.6359–0.6855) |
| **d** | | | |
| CT COVID-19 diagnosis | 0.9847 (0.9822–0.9877) | 0.9762 (0.9718–0.98110) | 0.91250 (0.8975–0.9301) |
| CXR COVID-19 diagnosis | 0.9527 (0.9474–0.9583) | 0.9623 (0.9570–0.9673) | 0.7155 (0.6918–0.7436) |
| Result-level fusion | 0.9894 (0.9873–0.9917) | 0.9469 (0.9399–0.9543) | 0.9503 (0.9371–0.9627) |
| **e** | | | |
| Pneumonia-or-non-pneumonia cohort | 0.9869 (0.9818–0.9993) | 0.9404 (0.9210–0.9756) | 1.0000 (1.0000–1.0000) |
| CAP-or-COVID-19 cohort | 0.9727 (0.9637–0.9825) | 0.9591 (0.9459–0.9767) | 0.9199 (0.8947–0.9512) |
| Influenza-or-COVID-19 cohort | 0.9585 (0.9413–0.9813) | 0.94961 (0.93333–1.0) | 0.8331 (0.7826–0.8846) |
| **f** | | | |
| Lung segmentation | 0.9255 (0.6018–0.9732) | 0.9660 (0.7553–0.9918) | 0.9956 (0.9787–0.9983) |
| **g** | | | |
| COVID-infectious slices locating | 0.9559 (0.9532–0.9586) | 0.8009 (0.79323–0.8094) | 0.9636 (0.9607–0.9666) |

a Metrics on test cohort. b Metrics on CC-CCII test cohort. c Metrics on MosMedData test cohort. d Metrics on comparison between CT and XCT on paired data of test cohort. e Metrics on three reader study tasks. f Lung segmentation performances. g COVID-infectious slices locating.

model was implemented in two-dimensional (2D) not only because it is easier to train within memory limit of common GPUs (usually 11 G), but also because slice-level diagnosis can be used for COVID-infectious slice locating. Other modules of our system are described in Supplementary Methods.

**Performances of diagnosis**. The trained AI system was evaluated on the test cohort. We used the receiver operating characteristic (ROC) curves to evaluate the diagnostic accuracy. On the test cohort, the ROC curve (Fig. 2a) showed AUC of four categories were respectively 0.9752 (for non-pneumonia), 0.9804 (for CAP), 0.9885 (for influenza), and 0.9745 (for COVID-19). Besides, sensitivity and specificity for COVID-19 were 0.8703 and 0.9660, and the multi-way AUC was 0.9781 (Table 2a). Our system showed good generalization ability with 0.9299 multi-way AUC on publicly available database CC-CCII (Fig. 2b and Table 2b), and 0.9325 AUC on MosMedData (Fig. 2c and Table 2c). The confusion matrix of four categories and PR curves of diagnosis of COVID-19 are shown in Fig. 3a–c, g. The decision curve analysis (DCA) for the AI system are presented in Fig. 3d–f, which indicated that the AI system added benefit when the threshold was within wide ranges of 0.03–0.87, 0.04–0.90, and 0.20–1.00 separately for COVID-19 diagnosis with negatives from non-pneumonia, CAP, and influenza. Lung segmentations were used as soft masks of images before images were fed into classification network, and the Dice index of our segmentation network was 92.55% (Table 2f).

COVID-infectious slice locating results are shown in Fig. 2d and Table 2g. Although with the same network structure as the slice diagnosis network, our experiments showed training COVID-infectious slice locating network, using normal and abnormal slices from COVID-19 subjects led to a much better

performance, with AUC of 0.9559, specificity of 0.9636, and sensitivity of 0.8009.

**Comparison of AI system to radiologists**. We conducted a reader study with five board-certified radiologists (average of 8-year clinical experience, range 5–11 years, Table 3a). All readers were asked to read independently without any information regarding whether the patient has been diagnosed with COVID-19.

Unlike Zhang et al.[24], we picked three different cohorts with different tasks for the reader study. The three cohorts with differential tasks were pneumonia vs. non-pneumonia, CAP vs. COVID-19, and influenza vs. COVID-19 (details described in "Methods"). The separate tasks helped us to analyze the COVID-19 distinguishing ability with different negative classes. Compared with human radiologists, the ROC curves in Fig. 2f–h and detailed metrics in Table 2e demonstrate that the AI system performed better than each radiologist at distinguishing CAP vs. COVID and influenza vs. COVID. This superior performance of the AI system can also be appreciated on a numerical level by the number of patients diagnosed correctly vs. incorrectly between the AI system and the radiologists (Table 3b, c). The AI system was only slightly worse at distinguishing pneumonia from non-pneumonia than radiologists.

Figure 4 shows some slices from error predictions of the AI system and human readers. Human readers tend to use some typical macro-level radiology features in diagnosis, so that the error predictions of human had atypical or unclear presentation. For example, the CT of COVID-19 in Fig. 4a has atypical density decrease (blue arrow) so that all readers classified it as CAP; the CT of influenza in Fig. 4b has multiple small patchy GGOs distributed around pleural and bronchial without pleural effusion,

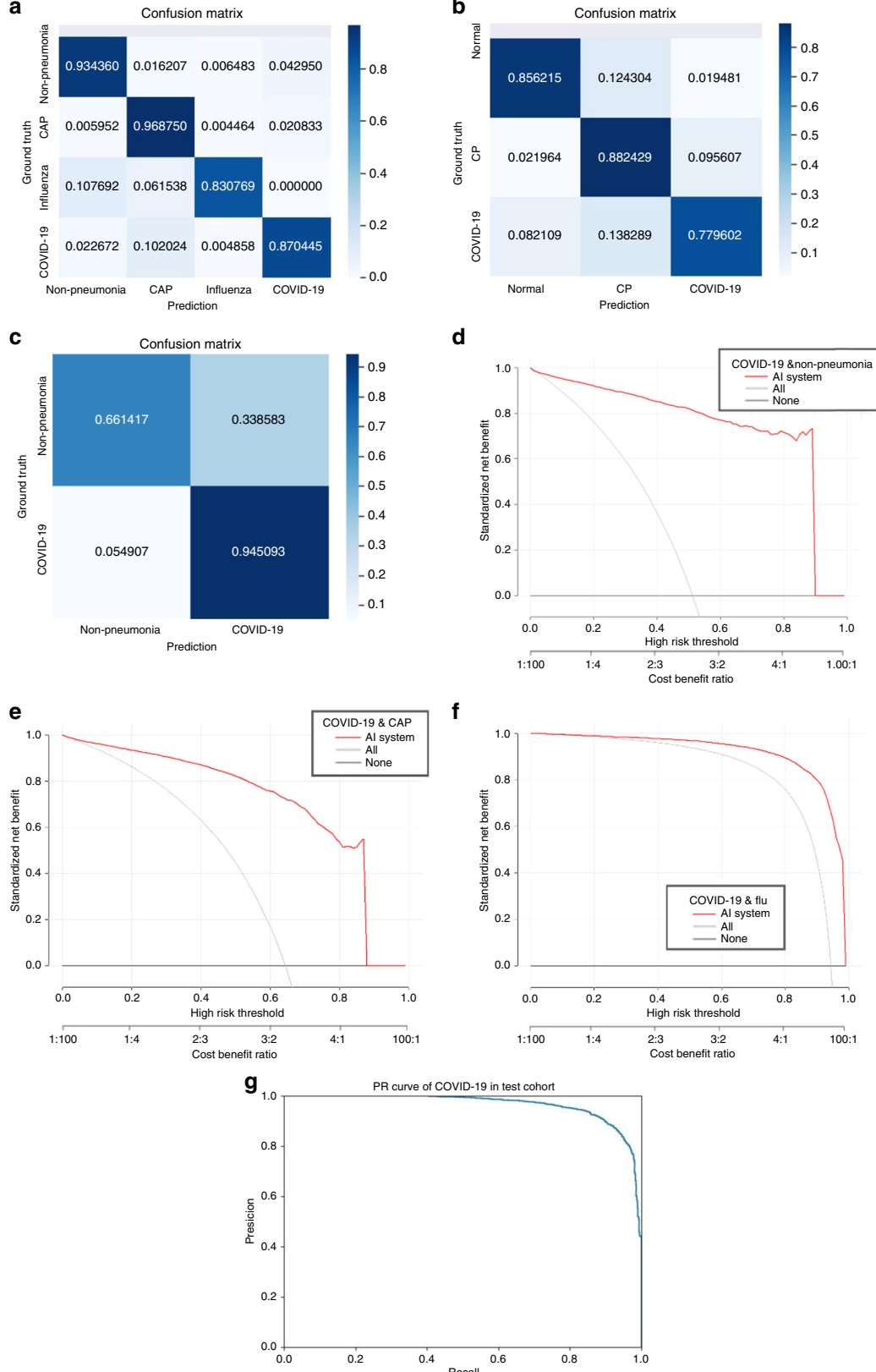

**Fig. 3 Performances of the AI system. a** Confusion matrix of the four diagnostic categories in the test cohort. **b** Confusion matrix of the CC-CCII test cohort. **c** Confusion matrix of the two diagnostic categories in MosMedData cohort. **d**–**f** DCA of the AI system in the test cohort for differential diagnosis of COVID-19, non-pneumonia, CAP, and influenza-A/B. **g** PR curves employed to assess the AI system performance on COVID-19 diagnosis.

**Table 3 Reader study statistics and results.**

**Reader information**

| | Reads per year | Years of experience | Reads on COVID-19 |
|---|---|---|---|
| **a** | | | |
| Reader 1 | 3000–4000 | 5 | 500–600 |
| Reader 2 | 4000–5000 | 10 | 500–600 |
| Reader 3 | 4000–5000 | 11 | 600–700 |
| Reader 4 | 3000–4000 | 8 | 600–700 |
| Reader 5 | 3000–4000 | 7 | 500–600 |

| | | | All reader | | Any reader | |
|---|---|---|---|---|---|---|
| | | | **Correct** | **Wrong** | **Correct** | **Wrong** |
| **b** | | | | | | |
| AI | Pneumonia-or-non-pneumonia | Correct | 97 | 0 | 95 | 2 |
| | | Wrong | 2 | 1 | 2 | 1 |
| | CAP-or-COVID | Correct | 92 | 2 | 69 | 23 |
| | | Wrong | 5 | 1 | 5 | 3 |
| | Influenza -or-COVID | Correct | 44 | 3 | 24 | 20 |
| | | Wrong | 3 | 0 | 3 | 3 |

| | | | Reader 1 | | Reader 2 | | Reader 3 | | Reader 4 | | Reader 5 | |
|---|---|---|---|---|---|---|---|---|---|---|---|---|
| | | | **Correct** | **Wrong** | **Correct** | **Wrong** | **Correct** | **Wrong** | **Correct** | **Wrong** | **Correct** | **Wrong** |
| **c** | | | | | | | | | | | | |
| AI | Pneumonia-or-non-pneumonia | Correct | 96 | 1 | 96 | 1 | 96 | 1 | 96 | 1 | 96 | 1 |
| | | Wrong | 2 | 1 | 2 | 1 | 2 | 1 | 2 | 1 | 2 | 1 |
| | CAP-or-COVID | Correct | 86 | 7 | 81 | 12 | 85 | 8 | 82 | 11 | 80 | 13 |
| | | Wrong | 6 | 1 | 4 | 3 | 6 | 1 | 5 | 2 | 5 | 2 |
| | Influenza-or-COVID | Correct | 33 | 11 | 31 | 13 | 34 | 10 | 35 | 9 | 35 | 9 |
| | | Wrong | 5 | 1 | 3 | 3 | 5 | 1 | 5 | 1 | 4 | 2 |

**a** Experience levels of the five radiologists involved in the reader study. **b** Comparison of diagnostic error between the AI system and human readers. In Any reader column, readers are deemed as wrong if anyone is wrong, otherwise deemed as correct. In All reader column, readers are deemed as wrong if all were wrong, otherwise deemed as correct. AI diagnosis was made at 0.5 threshold. **c** Comparison of diagnostic performances between the AI system and every human reader.

so all readers classified it as COVID-19. Both of these two cases were correctly diagnosed by AI. Figure 4c, d are two cases that AI misclassified, probably because lesions are too small or too close to lung margin.

**Subset analysis**. For an in-depth understanding of the AI system and characteristics of different populations with COVID-19, we evaluated the AI system on subsets of test cohort divided by gender, age, and stage and show them in Fig. 5a. The stage here was defined as rank of scans sorted by scanning date. To understand the cause for different diagnosis performances, we analyzed the COVID-infectious slice locating results in different subsets of COVID-19 patients (Fig. 5b). Subjects from LIDC–IDRI and Tianchi-Alibaba were anonymized so that the analysis were not done on them.

A subset of the patients in the database have multistage CTs. We compared the diagnostic performance of stage I and stage II and fusion of them. The experiment suggested that the performance of the AI system was independent of the progress of the disease because there were no significant qualitative differences between performances of different stages. We did not test more complex fusion methods which may overestimate the performance, since most non-pneumonia and CAP subjects have only one stage CT.

A subset of COVID-19 and CAP subjects in the database have localizer scans along with CT scans. The localizer scans of CTs are very similar to CXR, but typically noisier. We used this subset to study performances of CT vs. CXR. We developed a CNN-based classification algorithm to discriminate COVID-19 from CAP using these localizer scans (described in "Methods" and Supplementary Methods). Experiments on subjects with both types of data showed that CT-based system performed significantly better (Fig. 2d and Table 2d, $p < 0.001$). Representative examples are given in Fig. 6. CXR worked better than CTs only in a few cases (4 in 1022 cases), in which CTs had artifacts caused by breathing. But we also found some CTs with artifacts that were diagnosed correctly. To better understand the relationship between motion artifacts and diagnosis performance, further study with more specifically collected data is required (e.g., Fig. 6c). By fusion with CXR using simple score-level averaging, a slight benefit would be acquired compared to using CT only. Figure 6e showed an example that the result was corrected after fusion, while when CXR was in low quality, fusion can also bring in errors (e.g. Fig. 6f). A better fusion method might help achieve better performance, such as fusion in feature-level when training deep networks.

**Interpreting the AI system**. After proper training of the deep network, Guided gradient-weighted Class Activation Mapping (Guided Grad-CAM)[34] was exploited to explain the "black box" system and extract attentional areas, which is connected to the back end of the diagnostic model (Supplementary Fig. 4). Figure 7a shows some representative subjects for the visualization of Guided Grad-CAM to determine the attentional regions for each category. We used $t$-SNE[35] to map 2048-dimensional deep features to 2D plane (Fig. 7b), and the result showed that our model extracted powerful features to separate different categories in latent space.

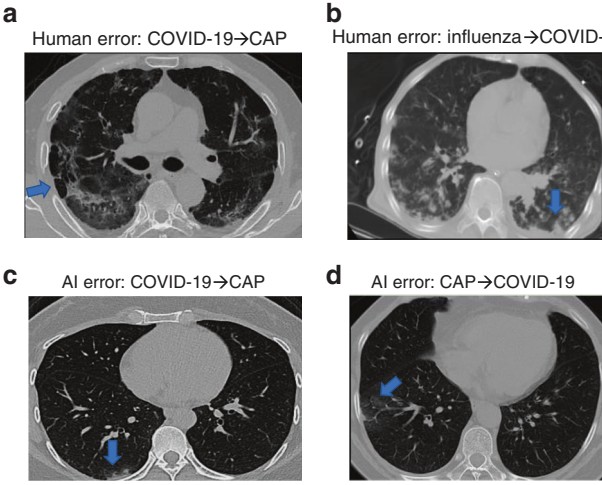

**Fig. 4 Examples of incorrect classifications by human readers and the AI system. a** CT image of a patient with COVID-19 that all five readers misclassified as CAP. **b** CT image of a patient with influenza that all five readers misclassified it to COVID-19. **c** CT image of a patient with COVID-19 that was misdiagnosed as CAP by AI. **d** CT image of a patient with CAP that AI diagnosed incorrectly as COVID-19. The blue arrows point out some infectious lesions.

We performed radiomics[36] feature extraction on these attentional regions, and obtained a total of 665-dimensional imaging features. The Least Absolute Shrinkage and Selection Algorithm (LASSO) were used to find the most discriminative 12 features in distinguishing COVID-19 from other pneumonia. Three additional features were also extracted for the attentional regions, distance feature, 2D margin fractal dimension, and 3D grayscale mesh fractal dimension ("Phenotype feature extraction" in "Methods"). The statistics of these features were consistent to previous literature[37] on the pathogenesis and morphology of COVID-19. The selected features were used to explain the imaging characteristics in CT ("Phenotype feature analysis" in "Methods"), and *t*-test and Kolmogorov–Smirnov (KS) test were performed to statistically analyze those features (Supplementary Figs. 7–9). According to the statistical analysis, we found that all features selected to distinguish CAP and COVID-19 were significant in *t*-test. KS test is used to test whether two groups of value come from two distributions, according to which wavelet-LL_firstorder_10Percentile and diagnostics_Image-original_Mean can help divide CAP and COVID-19 into two distributions. However, results for distinguishing influenza-A/B and COVID-19 were all not significant, which infers that radiomics features of attentional areas of influenza and COVID-19 have little difference. Cluster heatmaps after LASSO are also shown in Supplementary Figs. 7 and 8, according to which, we can know whether different classes were divided to different clusters by the selected features. Such figures allow us to understand the relationship between feature-level findings and image-level findings, which will be listed in "Discussion" and "Phenotype feature analysis" in "Methods".

## Discussion

In this study, we developed an AI system for diagnosis of COVID-19, CAP, influenza, and non-pneumonia. The multi-way AUC of the system is 0.9781 on our test cohort and 0.9299 on the publicly available CC-CCII database, and the AUC is 0.9325 on the publicly available MosMedData database. Even though CC-CCII does not provide the original CT volumes, our method can

still work well. Although subjects in MosMedData are from Russia, while the training data all come from China, the system seems to generalize well. Furthermore, in the reader study, the diagnostic accuracy of the AI system outperformed experienced radiologists in two tasks from the outbreak center, with AUC of 0.9869, 0.9727, and 0.9585 separately for pneumonia-or-non-pneumonia, CAP-or-COVID-19, and influenza-or-COVID-19 tasks. In the reader study, the average reading time of radiologists was 6.5 min, while that of AI system was 2.73 s, which can significantly improve the productivity of radiologists. Only in pneumonia-or-non-pneumonia cohort, the AI system worked slightly worse than human readers. In those more challenging tasks, the AI system worked better than human readers. For diagnosis between CAP and COVID-19, when the AI system misclassified, the radiologists were also wrong in 37.5% (3/8) of subjects (Table 3b), indicating that the diagnosis of these cases is challenging. And for influenza-A/B, that number is 50% (3/6). Meanwhile, we found that 88.5% (23/26) and 86.9% (20/23) of errors made by radiologists in those two tasks were correctly classified by the AI system. Differentiating COVID-19 from influenza is very difficult for human (76% accuracy averaged), but our AI system reached a good performance. It means that the AI system can be used as an effective independent reader to provide reference suggestions. Besides, with a highly sensitivity setting, it can screen out suspicious patients for radiologists to confirm; with a high accuracy setting, it can give possible diagnosis error warnings made by radiologists.

To further understand the performance of the AI system, we evaluated it on subsets divided by gender, age, and stage (Fig. 5a, b), which can assist decision-making in different populations. According to Fig. 5a, b, the number of infectious slices was changed with age, and the diagnostic performance was also changed with age. We concluded that young people might have less infectious area, resulting in lower diagnostic performance. The results on the subsets divided by gender showed little difference on infectious slices, but the average AUC for man was higher than that for women. This is consistent with the conclusion of Xiong et al.[38] that women have higher antiviral immunity than men, so that it was harder for AI to find out diagnostic clues in CTs of women. The results of different stages showed that the performance of the AI system had little correlation with the stages of CT scans. Fusion of stages can slightly improve the performances.

CXR is also considered as a possible way to diagnose COVID-19. According to Figs. 2d and 6, CXR had diagnostic value though it was generally not better than CT. As far as our survey, we are the first to compare CT and CXR performances in a paired cohort, which makes the comparison result more convincing. Importantly, the performance of CXR might be underestimated because localizer scans are in poor quality compared to normal CXR. Nevertheless, localizer CXR might be the best possible data to compare CT against CXR, since there is currently no dataset with paired CT and normal CXR captured in a very small interval.

According to Grad-CAM, we found that the AI system focused on different regions depending on the types of pneumonias. For CAPs, it generally ignored GGO which might also occur in COVID-19, and focused on effusion and consolidation adjacent to the pleura. On the other hand, the AI system focused on GGO rather than consolidation for most COVID-19 subjects. For influenza and COVID-19, Grad-CAM displays similar concerns, such as stripe consolidation and GGO, but the AI system is still able to distinguish the two correctly. We speculated that the AI system will focus on specific regions where other types of pneumonias may be rare. *T*-SNE clearly showed that deep features provided by our AI system can divide different types of

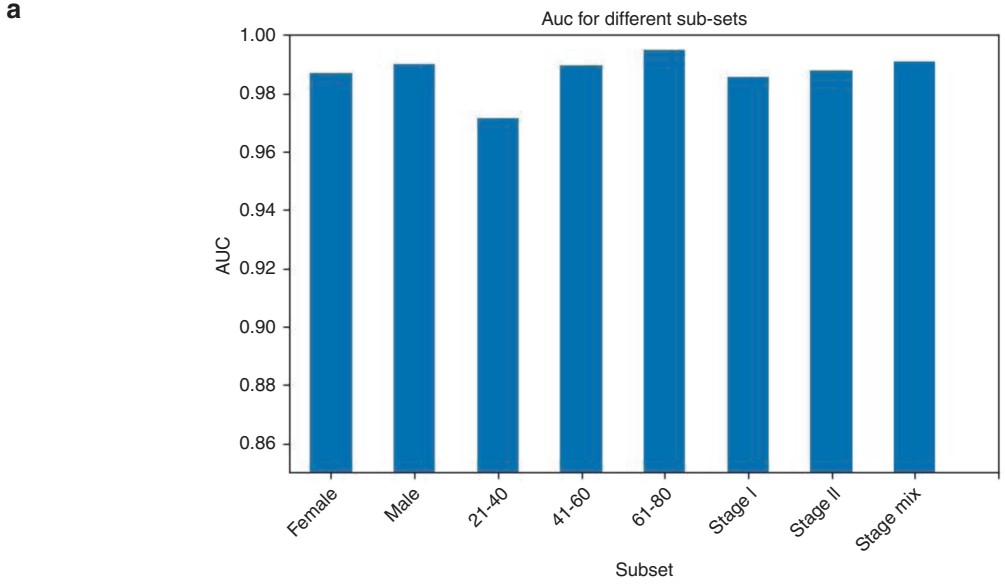

**Fig. 5 Statistics on different subsets of subjects in test cohort. a** AUC scores for each category on different subsets. **b** Distribution of ratio of COVID-infectious slices on different subsets of COVID-19 patients.

pneumonias into different clusters, as shown in Fig. 7b, and especially COVID-19 subjects were mapped to more than one clusters. By visualizing the raw images of the feature points, COVID-19 was found to have several types of presentations (left, upper, and right), which were enclosed by the red borders.

Samples in left cluster of COVID-19 were most in early and mild stage, which have small GGOs with nearly round shape. Samples in right cluster had larger lesions and some of them had crazy paving patterns. Fibration and consolidation could be found in the upper cluster whose size of lesion was generally between other

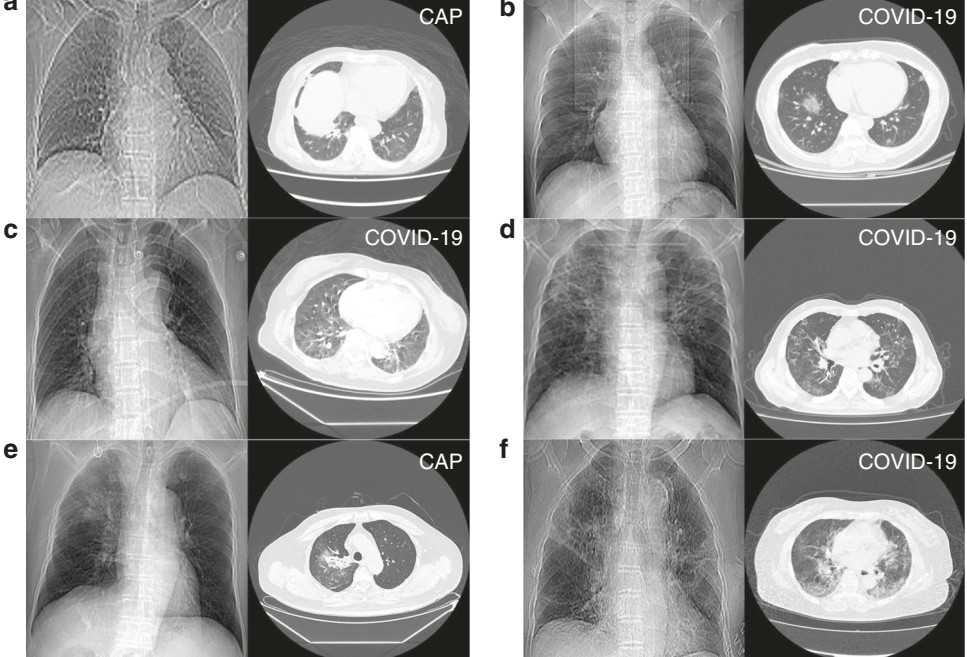

**Fig. 6 Representative diagnosis results of CT-based AI system and CXR-based AI system. a** An example that both CXR and CT were wrong. **b** An example that both CXR and CT were correct. **c** An example that CXR was correct, CT was wrong, and fusion result was wrong. **d** An example that CT was correct, CXR was wrong, and fusion result was correct. **e** An example that CXR was correct, CT was wrong, and fusion was correct. **f** A case that CXR was wrong, CT was correct, and fusion was wrong. Note: the true classes were annotated at the upper right of every images.

clusters. Although visualization by *t*-SNE was a conjecture for extracting features from the network, we can clearly find that patients of COVID-19 may be divided into different subclasses.

Further, we provided a visual interpretation of the system's decision by performing a radiomics analysis to obtain diagnostically relevant phenotypic characteristics of the attentional regions that are mapped to the original CT image. By visualizing the diagnostic results and the phenotype analysis, we found that the spatial distribution of the attentional region, morphology, and the texture within it are consistent with the characteristics of COVID-19 as reported in previous manual diagnosis studies[6,39], and we can make pathophysiological and anatomical speculations on the viral infection process ("Phenotype feature analysis" in "Methods").

There are still some drawbacks and future works of this research. Firstly, collecting more data on more subtypes of pneumonias or other lung deceases is useful for exploring AI system with higher diagnosis capability. Secondly, Guided Grad-CAM can only extract attention region rather than lesion segmentation, while phenotype feature analysis would better be done on accurate segmentation. Finally, constructing a large dataset with linked CT and clinical information, especially with information of underlying diseases, will enable additional analysis of the diagnosis system and development of more functionality, such as decease severity evaluation.

Overall, our AI system has been comprehensively validated on large multi-class datasets with higher diagnosis performance than human experts in diagnosing COVID-19. Unlike classical black box deep learning approaches, by visualizing the AI system and applying radiomics analysis, it can decode effective representation of COVID-19 on CT imaging, and potentially lead to the discovery of new biomarkers. Radiologists could perform an individualized diagnosis of COVID-19 with the AI system, adding new driving force for fighting the global spread of outbreak.

## Methods

**Development and validation datasets**. We retrospectively collected data from three centers in Wuhan, which are Wuhan Union Hospital, Western Campus of Wuhan Union Hospital, and Jianghan Mobile Cabin Hospital. The study has been approved by the institution review board of Wuhan Union Hospital, which is in charge of three centers. As a retrospective study, the need for informed consent was waived by the institutional review board. In total, 4260 CT scans (2529 COVID-19 scans, 1338 CAP scans, 135 influenza-A/B scans, and 258 normal scans) from 3177 subjects (1502 COVID-19 patients, 83 influenza-A/B patients, 1334 CAP patients except influenza, and 258 healthy subjects) were collected from multicenters (Table 1). CT volumes of COVID-19 patients were collected from February 5, 2020 to March 29, 2020, and all these patients were confirmed as COVID-19 by RT-PCR. Note that mild COVID-19 patients were also included because Jianghan Mobile Hospital was designated to treat mild patients, which increased the difficulty of diagnosis. For subjects with three or more scans, we excluded the last scan since the last ones might be rehabilitative. Table 1b shows characteristics of multi-scan data after exclusion. CTs of heathy subjects are from physical examinations of Union Hospital from January 2, 2020 to February 2, 2020. All these subjects were PCR negative and no pneumonia signs were found in their CTs according to CT diagnosis reports. CAP volumes were collected from January, 2019 to November, 2019. The CAP cases in our cohort were all nonviral pneumonias. Influenza-A/B volumes were collected from November, 2016 to November, 2019. All CAP and influenza subjects were retrospective and confirmed subjects who should not be COVID-19 according to study dates.

LIDC–IDRI and Tianchi-Alibaba are both databases for lung nodule detection with separately 1009 and 1200 scans available. All subjects of them suffered from benign or malignant lung nodules. Because nodules have totally different presentations, we set up a category "non-pneumonia" to cover both healthy subjects from Wuhan and subjects from LIDC–IDRI and Tianchi-Alibaba.

All above data (except CC-CCII and MosMedData) were randomly divided into training and test cohorts, with no overlapping subjects. The ratio of division is roughly 1:1. As a result, the cohort division is:

1. Training cohort: 2688 subjects (3263 scans) were assigned to training cohort which contained 1230 non-pneumonia, 666 CAP, 41 influenza-A/B, and 751 COVID-19 subjects. In this cohort, 198 CAP subjects (198 scans) and 468 COVID-19 subjects (725 scans) had localizer scans, which were considered as paired CXR and the CXRs were used to train CXR diagnosis network.

2. Test cohort: 2688 subjects (3199 scans) were assigned to test cohort, which contained 1229 non-pneumonia subjects, 668 CAP, 42 influenza-A/B, and 751 COVID-19 subjects. In this cohort, 220 CAP subjects (220 scans) and

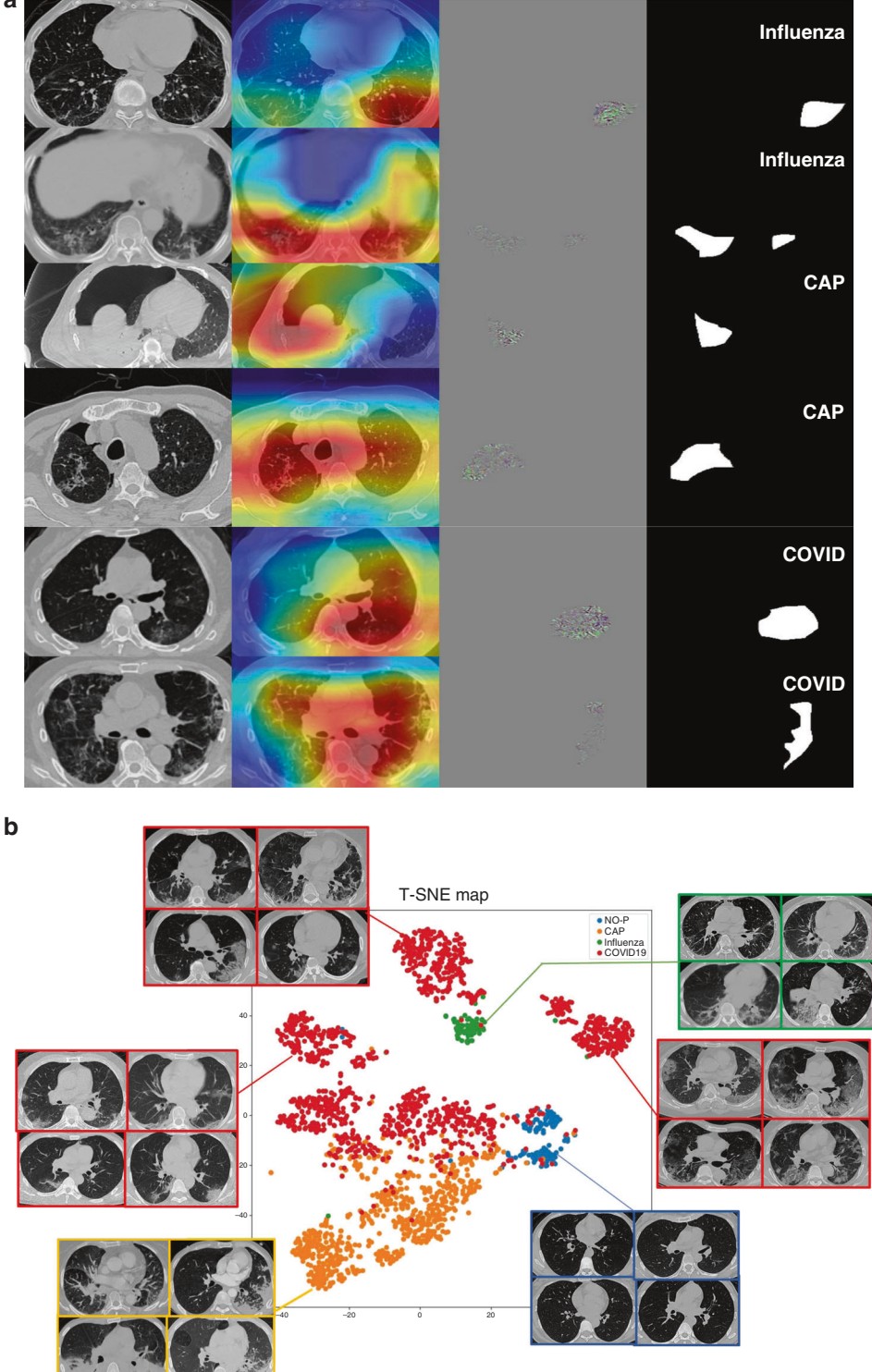

**Fig. 7 Interpreting the AI system. a** Visualizing feature maps via Grad-CAM. Notes: two slices of each of three representative subjects are used for the visualization of AI diagnosis. From left to right: original CT image; coarse-resolution attentional regions overlaid on CT image; high-resolution attentional regions with fine granularity; and binarized maps of region of attention obtained from Guided Grad-CAM. **b** Visualizing features via *t*-SNE.

469 COVID-19 subjects (802 scans) had localizer scans, which were considered as paired CXR. This part of data were used to evaluate CT performance compared with CXR.

3. CC-CCII test cohort: due to different category definition of CC-CCII database, we evaluated our system on it as an external independent test cohort. All its 2539 subjects (3784 scans) were used only in the test, we took the "common pneumonia" of CC-CCII as CAP together with influenza of

our system in experiments. Since CC-CCII database shares only processed image slices, from which the original CT values cannot be obtained, and the definition of categories is different, the diagnosis performance could be a little lower.

4. MosMedData test cohort: in this database, COVID-19 cases were separated into four categories: mild, moderate, severe, and critical. Since our AI system was trained to perform COVID-19 detection, all four categories were labeled

as COVID-19. As a result, there are 254 non-pneumonia and 856 COVID-19 cases in this cohort.

**Development of deep learning modules**. The lung segmentation module is implemented based on U-Net[40], which is a 2D semantic segmentation network. All CTs are in 3D, so we trained and tested the segmentation model slice by slice. The training slices were extracted from chest CTs in the training cohort and annotations of lung segmentation were obtained manually. Lung segmentations worked as masks and region boxes in diagnosis module, which were merged with raw images into a multichannel image as the input of classification network (described in Supplementary Methods).

The slice diagnosis module is a 2D classification deep network whose backbone is ResNet152 (ref. [41]). The parameters of ResNet152 were pretrained on a huge dataset ImageNet[9] for better and faster convergence. We tested a 3D classification network, but this 2D scheme showed much better performance and 3D network might not work if GPU memory is limited (even 11 G memory might not be able to process one volume of high resolution). In order to eliminate the influence of different scanners and factors on the diagnosis, the inputs of slice diagnosis module were lung-masked slices, which had been cropped out along their lung bounding boxes, and segmentation mask. The outputs of the classification network were four scores, respectively, representing confidence levels of being four categories (three categories for CC-CCII database). Slices for training this network were extracted from training scans, and the extraction process is explained in Supplementary Methods.

A task-specific fusion block is used to get a volume/case-level prediction from slice-level results. Because one volume is regarded as pneumonia infected if any one of its slices is diagnosed as infected, the fused scores of three pneumonia classes (CAP, Influenza, and COVID-19) were obtained by averaging the top-$K$ highest scores of each class ($K$ was 3, chosen by experiments) in all slices and the non-pneumonia score is obtained by averaging all slices. More details of the hyper-parameter choice are stated in Supplementary Methods. If the task is specific to distinguish non-pneumonia and pneumonia subjects, all pneumonia scores of a slice will be summed up. If the task is specific to distinguish COVID-19 from other pneumonia, the scores of other classes will be muted in the fusion block. To measure the diagnostic performance, AUC, sensitivity, and specificity (with default threshold 0.5) are computed on test cohort.

COVID-infectious slice locating module has the same structure as the slice diagnosis module, but it was trained and evaluated only on a set of COVID-19-positive subjects whose slices with lesions have been marked manually. All training samples of this module came from training cohort, and evaluation samples came from test cohort.

We used Guided Grad-CAM to obtain attentional regions as interpreting block of our system. Guided Grad-CAM has the advantage that it not only generates a heat map to locate the relevant area, but also produces a coarse localization map highlighting the important regions in the image for predicting the result. Guided Grad-CAM may also give more detailed diagnosis suggestions in addition to classification results. Also, the attentional regions were used in latter feature extraction and analysis to get more detailed information about lesion areas. We extracted region of attention by binarizing the output of Guided Grad-CAM followed by some morphological operations.

T-SNE was done based on features from slice diagnosis module, which are obtained by max-pooling feature maps of all slices before the last fully connected layer. Therefore, every CT volume was mapped to a 2048-D latent feature vector for $t$-SNE analysis.

All the deep learning blocks were implemented using PyTorch (version 1.3.1)[42]. T-SNE was implemented using scikit-learn (version 0.22) package. Numpy (version 1.15.3) and scipy (version 1.3.3) were also used in analysis.

**Reader study**. In order to analyze the performance, we set up three different reader study cohorts for different diagnosis tasks, which were acquired by randomly selecting subjects from test cohort:

1. Pneumonia-or-non-pneumonia cohort: 100 subjects (50 non-pneumonia subjects, 25 CAP subjects, and 25 COVID-19 subjects from three centers) were assigned. This cohort was used to compare results of the AI system with radiologists in diagnosis of pneumonia.
2. CAP-or-COVID-19 cohort: 100 subjects (50 CAP subjects and 50 COVID-19 subjects from three centers) were assigned. This cohort was used to compare results of the AI system with radiologists in distinguishing COVID-19 from CAP.
3. Influenza-or-COVID-19 cohort: 50 subjects (20 influenza-A/B subjects and 30 COVID-19 subjects from three centers) were assigned. This cohort was used to compare results of the AI system with radiologists in distinguishing COVID-19 from influenza-A/B.

We invited five experienced radiologists in our experiments from the radiology department of Wuhan Union Hospital, which is in the center of the epidemic area with the most patients in this outbreak in China. They all have read over 400 CTs of COVID-19 in the past 3 months. Five radiologists had an average of 8 years of

clinical experience in the imaging diagnosis of pulmonary diseases, as detailed in Table 3a.

The human radiologists were aware of the tasks and the possible classes in reader cohort when reading. For example, they were informed that only CAP and COVID-19 subjects were collected in CAP-or-COVID cohort. Besides, readers can choose any window of gray value, and zoom in or out when reading CT volumes using Slicer 4.10.2 software while our system used fixed size resample images ($224 \times 224 \times 35$) with fixed gray value window ($-1200, 700$) for all volumes.

**Attentional region extraction**. To analyze the differences of imaging phenotype between different pneumonia, features were extracted in the attentional regions determined by binarizing Grad-CAM maps followed by morphological processing. We only kept the regions with valid size (>200 pixels within margin of lung mask). We extracted attentional regions only in pneumonia subjects in test cohort.

**Phenotype feature extraction**. Radiomics features widely used in tumor diagnosis were extracted. These features were composed of different image transforms and feature matrix calculations. We adopted three image transforms: original image, transformed image by Laplacian of Gaussian operator, and transformed image by wavelet. For each image after the operation of a transform, six series of features are extracted, including first order features, Gray Level Co-occurrence Matrix, Gray Level Size Zone Matrix, Gray Level Run Length Matrix, Neighboring Gray Tone Difference Matrix, and Gray Level Dependence Matrix. Radiomics analysis was performed using python version 3.6 and the pyradiomics (version 2.2.0) package[36].

The distance feature was defined as the distance between the center of gravity of the region of interest (obtained by Grad-CAM) and the edge of the lung (obtained by lung segmentation results). Besides, 2D contour fractal dimension and 3D grayscale mesh fractal dimension of the attentional region were extracted. The fractal dimension describes the degree of curvature of a curve or surface.

LASSO logistic regression model was used to choose most discriminative features in all extracted ones. LASSO analysis was performed using python version 3.6 and the scikit-learn package.

**Phenotype feature analysis**. For distance feature (Supplementary Fig. 9a), there were two peaks of distances of COVID-19 that were generally 0–30 pixels (2.5 mm/pixel) from the pleura and a little amount of which were >100 pixels. That is different with other distributions that most of CAPs were not >40 pixels to pleura. Distribution of influenza-A/B was flatter, which is consistent with anatomical findings on COVID-19. We found that the distribution was consistent with pathological study. When the SARS-Cov2 virus is inhaled through the airways, it mainly invades the deep bronchioles, causing inflammation of the bronchioles, and their surroundings to damage alveolar[43,44]. These areas have well-established immune system and well-developed pulmonary lobules, leading to a strong inflammatory response[45,46]. Secondly, because region of attention obtained by Grad-CAM could not delineate the lesion accurately, there were little differences among three types of pneumonia in fractal dimension (Supplementary Fig. 9b). Thirdly, COVID-19 was a little fickler in gray value compared to others with higher 3D fractal dimension, while CAP had two peaks in this feature. According to $p$-value, distance feature and 3D fractal dimension can help distinguish CAP and COVID-19, whereas the 2D fractal dimension showed little information (Supplementary Fig. 9c). No significant difference could be found between influenza-A/B and COVID-19 on those features and according to KS test only distance feature is significative to distinguish CAP and COVID-19.

According to all selected features, we can describe in depth the relationship between the medical findings and typical patterns of COVID-19 (some examples are shown in Supplementary Fig. 5). (I) Halo and antihalo pattern: the halo pattern was speculated to be that the lesions (mainly the central node of the lobular) infiltrated into the surrounding interstitium and developed the aggregation of inflammatory cells in the interstitium. Antihalo pattern was ground glass shadow surrounded by the high-density consolidation. The reasons why this sign appeared may be that the inflammatory repair was dominated by the edge, leading to the formation of a band shadow tending to consolidation at the edge, while the central repair was relatively slow. (II) Pleural parallel signs: the formation mechanism was speculated as follows: when the COVID-19 invaded the interstitium around the alveoli, the lymphatic return direction was subpleural and interlobular septa, and diffused into pleural side and bilateral interlobular septum[47]. Because of the limitation of the pleura at the distal end, the lymph can only cling to the pleura and spread along the reticular structure of the interlobular septal margin on both sides. In addition, the fusion of the subpleural lesions resulted in the long axis of the lesions parallel to the pleura. (III) Vascular thickening: it was consistent with the rules of inflammation production. The inflammatory increased vascular permeability, caused telangiectasia, further caused pulmonary artery thickening[46,48]. (IV) The fine mesh feature of large area: the COVID-19 mainly invaded the interstitium in the lobules, so it appeared as confluent fine mesh (crazy paving). (V) The density-

increased GGO. This kind of GGO was transforming to consolidation. The consolidation edges were flat or contracted, and fiber strands appeared. Compared with COVID-19, influenza showed higher density on the lesion area, which can be inferred that they caused slightly less alveolitis and GGO patterns. As a result of bronchiolitis, influenza virus pneumonia was more likely to form tree-bud signs, and occasionally hyperlucent lung.

**Reporting summary**. Further information on research design is available in the Nature Research Reporting Summary linked to this article.

## Data availability

Four databases in our experiments are publicly available. LIDC–IDRI database can be accessed at https://wiki.cancerimagingarchive.net/display/Public/LIDC-IDRI. Tianchi-Alibaba database can be accessed at the webpage (https://tianchi.aliyun.com/competition/entrance/231601/information?lang=en-us) of the challenge after registration. CC-CCII database can be accessed at http://ncov-ai.big.ac.cn/download. MosMedData database is available at https://mosmed.ai/en/. The datasets from Wuhan Union Hospital, Western Campus of Wuhan Union Hospital, and Jianghan Mobile Cabin Hospital were used under the license of the current study from Union Hospital, Tongji Medical College, Huazhong University of Science and Technology, Wuhan, China (2020/0030), so they are not publicly available. Interested readers may contact Heshui Shi for further information about these datasets. Source data are provided with this paper.

## Code availability

To help combat with COVID-19 which is spreading around the world, we decided to opensource our AI system to facilitate testing and further development. The project can be forked from https://github.com/ChenWWWeixiang/diagnosis_covid19.

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

## Acknowledgements
We would like to acknowledge the radiologists participating the reader study. This study was supported by Zhejiang University special scientific research fund for COVID-19 prevention and control.

## Author contributions
C.J., H.S., and J.F. contributed to the conception of the study; C.J., W.C., Z.X., Z.T., and J.F. designed the algorithms; Y.C., X.Z., L.D., and C.Z. contributed to acquisition and annotation of the data; H.S. and J.Z. contributed to analysis and interpretation of the data; and C.J., W.C., and J.F. contributed to drafting and revising the manuscript.

## Competing interests
The authors declare no competing interests.
