## [Peer Review File · Nature Communications]

Reviewers' Comments:

Reviewer #1:

Remarks to the Author:

1. This manuscript would benefit by being reviewed and amended by a native English speaker
2. I am not sure that some of the statements in the introduction are accurate or supported:
A - CT access is relatively limited in comparison to the CXR
B - I am unsure of the cost of PCR v CT
D - It does not appear to be difficult to train radiologists to identify COVID-19 on CT
E - I don't know the shortfall of radiologists able to interpret COVID-19 CT scans
3. The results are excellent but the AI algorithm performed less well on younger patients suggesting that identification of disease relates to its severity. I do not know whether the authors have the ability to sub classify the patients COVID disease severity and report this
4. The majority of patients included will have been admitted, as patients with mild disease are both often not admitted and do not undergo CT, so this should be discussed
5. The comparison with the CXR is interesting, and its performance. The comment on breath hold suggests that the algorithm may not have been usable in a percentage of patients and this should be made clear
6. It is difficult to extrapolate from the data presented whether patients with other diseases were included ie pulmonary fibrosis and cardiac failure, and whether the algorithm performed less well in these patients. A comment on this would be helpful.

Reviewer #2:

Remarks to the Author:

Thank you for the opportunity to review this timely and important manuscript titled "Development and Evaluation of an AI System for COVID-19 Diagnosis." The objective of the study conducted by Dr. Jin and colleagues was to develop an artificial intelligence system based on Chest CT scans to accurately and quickly diagnose COVID-19. The authors successfully develop an AI system that distinguishes COVID-19, Influenza, community acquired pneumonia, and healthy controls rapidly with high diagnostic accuracy. They also compared their findings to experienced radiologists, demonstrating the AI system was as good as radiologists and much faster. The manuscript has potential clinical relevance and the authors should be commended for their work. The authors have significantly improved their manuscript from the original submission. Most importantly, they now determine the diagnostic accuracy of their AI system in patients with diseases that are similar to COVID-19 (influenza, CAP).

1. In the manuscript, the authors refer to some subjects having multiple CTs. Did the authors include multiple CTs from single patients in the AI system development and testing?
2. What is meant by stage? Severity of pneumonia? Of respiratory failure?

Reviewer #4:

Remarks to the Author:

The manuscript proposes an artificial intelligence (AI) system for fast COVID-19 detection and performs extensive statistical analysis of CTs of COVID-19 based on the AI system. It develops and evaluates the system on a large dataset with more than 10 thousand CT volumes from COVID-19, influenza-A/B, non-viral community acquired pneumonia (CAP) and non-pneumonia subjects. Extensive experiments demonstrate the effectiveness of the proposed method.

The paper is well-written with extensive experiments. However, there are lots of AI systems for lung CT and many important information are missing. The paper can be improved by addressing the following concerns.

1 Page 3, line 31, "contourites" might be "countries"?

2 I do not agree with the novelty of using paired CT and CXR. CXR can be fully constructed by CT. There is no reason why the system with paired CT and CXR is better than CT. And system with additional CXR costs more money.

3 There are a lot of work demonstrating AI-based lung disease diagnosis system is comparable or better than experienced radiologists. The novelty of this manuscript related with this part is limited. And the manuscript does not refer previous related work properly.

4 In figure 2 and other figures, please zoom in the up-left corner of the ROC so that reader can clearly find the difference.

5 The manuscript uses a private dataset to evaluate the performance. How to reproduce the results is a question. Results on a public dataset can be used to reduce reader's concern.

6 In Page 26, the manuscript uses segmentation as pre-processing. What is the performance of segmentation? How to handle the failed segmentation?

7 In Page 26, line 431, 432, did the manuscript use other networks instead of ResNet152? It is better for the manuscript to provide a network architecture so that reader can understand much better.

8 In Page 27, I do not think averaging top-K is reasonable. Let us say, the maximum probability is 0.9, and the second, third maximum probability is 0.2, 0.1. The average is 0.4, which means negative. However, it is positive because it has slice of 0.9 predicted probability. Also, how to choose K?

9 In page 27, line 447, 448, "all scores of a slice will be summed up". It seems it might be "average"?

10 In Page 27, line 460, it is difficult to understand the "secondary output". Adding a network structure might help the understanding.

11 In Page 40, 42, what is the purpose of these figures? What is the reason for the result? Adding more information might help readers to understand.

12 In lines 416-422, the manuscript retrains the model when performing "external validation", which might be inappropriate. The "external validation" is used to validate the model on a holdout dataset and retraining breaks the independent of the model and the holdout test dataset.

13 From the manuscript, it uses multi-way classification. The network structure should be explicitly inserted so that readers can understand it better. Also, the manuscript should report the all the related performance and the confusion matrix for these classes.

Reviewer #1:

1. This manuscript would benefit by being reviewed and amended by a native English speaker.

Thanks for the advice. We have requested a native English speaker to proofread the manuscript.

2. I am not sure that some of the statements in the introduction are accurate or supported:

Thanks for the comments. The previous statements are valid at a certain period in certain countries. We have revised the statements to ensure that they are valid in general situation.

A - CT access is relatively limited in comparison to the CXR.

We agree to this comment. We have added the following sentence: “Chest imaging, especially CT, can show early lesions in the lung and, if diagnosed by experienced radiologists, can achieve high sensitivity. In addition, chest imaging, especially CXR, are widely available and economic.”

B - I am unsure of the cost of PCR v CT.

The price and availability of PCR vary greatly from time to time and place to place, and the cost of CT varies greatly from country to country, making it difficult to compare. But CT is still an important tool for diagnosis and assessment of COVID-19 even when PCR is cheap and available. For PCR and CT, we have changed the statement from “kits of RT-PCR are in short of supply in many areas.” to “the supply of kits of RT-PCR varies from place to place that some developing countries are in short supply of it.”

D - It does not appear to be difficult to train radiologists to identify COVID-19 on CT.

It is not difficult for radiologists to distinguish COVID-19 patients from healthy subjects based on CT. But distinguishing COVID-19 from other similar pneumonias is not easy for radiologists. Especially for influenza, which is also viral pneumonia, even well-trained radiologists in epidemic center can make many mistakes in our reader study.

E - I don't know the shortfall of radiologists able to interpret COVID-19 CT scans.

Thanks for the comment. We have changed the statement from “The current outbreak of COVID-19 is worldwide, and the shortage of specialist radiologists threatens the availability and adequacy of screening services for COVID-19 in affected areas” to “Radiologists need to accumulate a lot of CT diagnostic experience to achieve a high diagnostic performance, especially in differentiating similar deceases. COVID-19 can still be a threat for a long time since the situations in some countries are not optimistic. If COVID-19 and influenza were to break out together, which is possible, CT diagnosis workload would likely be far beyond the number of qualified radiologists.”

3. The results are excellent but the AI algorithm performed less well on younger patients suggesting that identification of disease relates to its severity. I do not know whether the authors have the ability to sub classify the patients COVID disease severity and report this.

Thanks for raising this question. It is a good idea to extend the COVID-19 detection algorithm to estimate the disease severity. COVID-19 severity is clinically defined based on symptoms, oxygenation index, serological index and so on. Our algorithm can be extended to estimate severity, as long as the groundtruth is available to us. Unfortunately, we do not have the groundtruth of disease severity up to now.

4. The majority of patients included will have been admitted, as patients with mild disease are both often not admitted and do not undergo CT, so this should be discussed.

According to COVID-19 management policy of China, all confirmed patients

including mild stage patients should be admitted in hospitals. In addition, both PCR and CT are used for confirmation of COVID-19 in China. In fact, most subjects from Jiangnan Mobile Cabin Hospital, one of our three centers, were mild stage patients. We added the following sentence: “Mild subjects were also included because Jiangnan Mobile Hospital were designated to treat mild subjects, which increased the difficulty of diagnosis.”

5. The comparison with the CXR is interesting, and its performance. The comment on breath hold suggests that the algorithm may not have been usable in a percentage of patients and this should be made clear.

We have added the following sentence: “CXR worked better than CTs in a few cases (4 in 1022 cases) in which CTs had artifacts caused by breathing. But we also found some cases with artifacts that were diagnosed correctly. To better understand relationship between motion artifacts and diagnosis performance, further study with more specifically collected data is required.”

6. It is difficult to extrapolate from the data presented whether patients with other diseases were included ie pulmonary fibrosis and cardiac failure, and whether the algorithm performed less well in these patients. A comment on this would be helpful.

Thanks for the advice. Underlying diseases in addition to COVID-19 may influence diagnosis accuracy. In our dataset, certain patients might have other diseases, but we don't have this information. Analysis of diagnostic performances of patients with underlying diseases will be an interesting research direction.

Reviewer #2:

Thank you for the opportunity to review this timely and important manuscript titled “Development and Evaluation of an AI System for COVID-19 Diagnosis.” The objective of the study conducted by Dr. Jin and colleagues was to develop an artificial intelligence system based on Chest CT scans to accurately and quickly diagnose COVID-19. The authors successfully develop an AI system that distinguishes COVID-19, Influenza, community acquired pneumonia, and healthy controls rapidly with high diagnostic accuracy. They also compared their findings to experienced radiologists, demonstrating the AI system was as good as radiologists and much faster. The manuscript has potential clinical relevance and the authors should be commended for their work. The authors have significantly improved their manuscript from the original submission. Most importantly, they now determine the diagnostic accuracy of their AI system in patients with diseases that are similar to COVID-19 (influenza, CAP).

1. In the manuscript, the authors refer to some subjects having multiple CTs. Did the authors include multiple CTs from single patients in the AI system development and testing?

Thanks for the question. We have stated in both **Datasets for System Development and Evaluation** section in main text and **Development and Validation**

Datasets section in methods that our division was done on subject-level. Multiple CTs from single patients were not used in the AI system development and testing separately.

2. What is meant by stage? Severity of pneumonia? Of respiratory failure?

Thanks for pointing out that the word “stage” may be confusing. In our work, the “stage” means the order of CT scans sorted by time. Stage I means the first CT scan and the stage II means the second CT scan. We have defined them clearly in the revision.

Reviewer #4:

The manuscript proposes an artificial intelligence (AI) system for fast COVID-19 detection and performs extensive statistical analysis of CTs of COVID-19 based on the AI system. It develops and evaluates the system on a large dataset with more than 10 thousand CT volumes from COVID-19, influenza-A/B, non- viral community acquired pneumonia (CAP) and non-pneumonia subjects. Extensive experiments demonstrate the effectiveness of the proposed method.

The paper is well-written with extensive experiments. However, there are lots of AI systems for lung CT and many important information are missing. The paper can be improved by addressing the following concerns.

1. Page 3, line 31, “contourites” might be “countries”?

Thanks for pointing out and it is fixed.

2. I do not agree with the novelty of using paired CT and CXR. CXR can be fully constructed by CT. There is no reason why the system with paired CT and CXR is better than CT. And system with additional CXR costs more money.

In the perfect situation, CT can fully construct CXR, but in some cases the two modalities may have complementary information. Fusion of them can improve a little performance because of two reasons. First, the CXRs have no moving artifacts which are present in a few CTs. Second, 2D views can be regarded as an attentional area of 3D volumes, and combining 3D volumes with their attentional areas can possibly help improving feature selection and decision making of AI system.

Besides, our CXRs are not regular CXRs, but directly taken from localizer CXR of CT scans, which are always performed during CT scan to make sure that the scanned areas are accurate. So, there is no additional cost for obtaining CXR in our case. It will be better to compare regular CXR and CT. However, such study is not feasible now since subjects will receive more unnecessary radiations.

3. There are a lot of work demonstrating AI-based lung disease diagnosis system is comparable

or better than experienced radiologists. The novelty of this manuscript related with this part is limited. And the manuscript does not refer previous related work properly.

Thanks for the suggestion. We have added some representative publications about AI-based diagnosis, including lung nodule detection¹⁻³, tuberculosis diagnosis⁴⁻⁵ and lung cancer screening⁶. Different from lung nodule detection, which is usually viewed as an object detection problem, COVID-19 diagnosis needs to differentiate subtle differences between different types of pneumonias. Compared to lung cancer diagnosis or tuberculosis diagnosis, differentiating COVID-19 from other pneumonias has unique difficulty, high similarity of pneumonias of different types, especially in early stage. Hence developing AI diagnosis system specific to COVID-19 is necessary. As to other publications on AI based COVID-19 diagnosis, we have compared with them and referred to related work in our previous manuscript. (Paragraph 3, Main text)

4. In figure 2 and other figures, please zoom in the up-left corner of the ROC so that reader can clearly find the difference.

Good idea. We have changed it.

5. The manuscript uses a private dataset to evaluate the performance. How to reproduce the results is a question. Results on a public dataset can be used to reduce reader's concern.

It is useful to test on public datasets to confirm the reproducibility. Since COVID-19 is a new disease and is still outbreaking, we found only one publicly available CT dataset called CC-CCII (<http://ncov-ai.big.ac.cn/download?lang=en>), which had been used in our experiments. Since CC-CCII is a multi-center database and our database is also multi-center, we believe the performance evaluation is reliable. Besides, we have made our code publicly available, including preprocessing, training and test scripts, so that everyone can reproduce this work and evaluate it on their own databases.

6. In Page 26, the manuscript uses segmentation as pre-processing. What is the performance of segmentation? How to handle the failed segmentation?

Thanks for the advice. We have reported the performance of segmentation in the revised manuscript. In this study, the lung segmentation accuracy achieved 92.5% of Dice coefficient, which is sufficiently accurate for classification. In addition, the data were fed into the network in a multi-channel pattern, including a segmented lung field and corresponding entire CT slice. Such a soft-mask design is insensitive to small mistakes in segmentation. (Paragraph 1, Development of Deep Learning Modules, Methods).

7. In Page 26, line 431, 432, did the manuscript use other networks instead of ResNet152? It is better for the manuscript to provide a network architecture so that reader can understand much better.

We had tried other classic structures including different visions of Resnet, VGG, Inception and Densenet. Based on the performance on training set, we chose

Resnet152. The structure of Resnet152 is available on supplementary method and we have changed to a more reader friendly style of structure figure in our revision.

8. In Page 27, I do not think averaging top-K is reasonable. Let us say, the maximum probability is 0.9, and the second, third maximum probability is 0.2, 0.1. The average is 0.4, which means negative. However, it is positive because it has slice of 0.9 predicted probability. Also, how to choose K?

Thanks for raising this question. The choice of K is a tradeoff between sensitivity and specificity. For example, COVID-19 cases (even for mild cases) usually have multiple slices with typical COVID-19 patterns, while some CTs of other deceases might have a few slices with similar COVID-19 patterns. Top-K average can bring down these false positives if the K is chosen properly. In order to get a proper choice of K, we performed cross-validation on training cohort, which was not reported in previous version. This experiment has been reported in supplementary of the revised manuscript.

9. In page 27, line 447, 448, "all scores of a slice will be summed up". It seems it might be "average"?

Thanks for the question. Sum is used because we want to combine the probabilities of all other kinds of pneumonias.

10. In Page 27, line 460, it is difficult to understand the "secondary output". Adding a network structure might help the understanding.

It is a good point that GradCAM cannot be regarded as an output, but a tool to visualize deep networks. We have changed the statement in method parts and presented the structure of GradCAM in supplementary.

11. In Page 40, 42, what is the purpose of these figures? What is the reason for the result? Adding more information might help readers to understand.

Thank you for raising this question. These figures really need more detailed explanation. We have provided more information and analyze these results in Extended Data Figure 2. In Extended Data Figure 4 and 5, we tried to show coefficients and the significant degrees of features chosen by LASSO. More descriptions are added in revision.

12. In lines 416-422, the manuscript retrains the model when performing "external validation", which might be inappropriate. The "external validation" is used to validate the model on a holdout dataset and retraining breaks the independent of the model and the holdout test dataset.

We did not directly validate our model on the CC-CCII database because this database only offers processed image data (.png files) instead of raw data (.dcm files) (<http://ncov-ai.big.ac.cn/download?lang=en>). The raw CT value of this database, which is important for differentiating lesions from similar structures, is not available to us. The CTs of this database have been preprocessed by manually defined

WindowCenters and WindowWidthes, which could be different for different CTs because they were obtained by different doctors, could not be reverted to raw CT data. We contacted the authors of CC-CCII database for raw data, but they could not offer us that. Therefore, we had to preprocess our CT data to be as much as similar to the CC-CCII database, which can almost overcome the domain bias caused by unknown preprocessing parameters. Even so, our model still got good results with 0.9299 AUC on CC-CCII.

13. From the manuscript, it uses multi-way classification. The network structure should be explicitly inserted so that readers can understand it better. Also, the manuscript should report the all the related performance and the confusion matrix for these classes.

It is a good advice. The related performance and the confusion matrix are now listed in Extended Data Table 2, Extended Data Figure 1, and network structures are listed in supplementary since the network is a classic design of Resnet152. We also renew the workflow figure for readers to understand it more easily.

1. Xie H, Yang D, Sun N, et al. Automated pulmonary nodule detection in CT images using deep convolutional neural networks. Pattern Recognition. 2019.
2. Liao F, Liang M, Li Z, et al. Evaluate the Malignancy of Pulmonary Nodules Using the 3-D Deep Leaky Noisy-OR Network. IEEE Trans. Neural Networks Learn. Syst. 2019.
3. Zhu W, Liu C, Fan W, et al. DeepLung: Deep 3D dual path nets for automated pulmonary nodule detection and classification. In: Proceedings - 2018 IEEE Winter Conference on Applications of Computer Vision, WACV 2018.; 2018.
4. Nash, M. et al. Deep learning, computer-aided radiography reading for tuberculosis: a diagnostic accuracy study from a tertiary hospital in India. Scientific reports 10, 1-10 (2020).
5. Qin, Z. Z. et al. Using artificial intelligence to read chest radiographs for tuberculosis detection: A multi-site evaluation of the diagnostic accuracy of three deep learning systems. Scientific reports 9, 1-10 (2019).
6. Ardila, D. et al. End-to-end lung cancer screening with three-dimensional deep

learning on low-dose chest computed tomography. *Nature medicine* 25, 954-961
(2019).

Reviewers' Comments:

Reviewer #1:

None

Reviewer #4:

Remarks to the Author:

The authors have addressed most of my comments.

The demographic characteristics are pretty important in clinical practice. Since many AI systems have exposed to be biased to some groups of subjects, it would be better if the system is validated to be fair for different groups of patients. In the experiments, the fairness of the system might be needed.

Reviewer #4 (Remarks to the Author):

The authors have addressed most of my comments. The demographic characteristics are pretty important in clinical practice. Since many AI systems have exposed to be biased to some groups of subjects, it would be better if the system is validated to be fair for different groups of patients. In the experiments, the fairness of the system might be needed.

Response:

Thanks for this suggestion. In our previous study, we have already evaluated our system on subsets divided by age and gender. As shown in Figure 5, there is no obvious difference among subsets of different ages and genders. Considering the comment of reviewer #4, we choose to include a new publicly available COVID-19 database, MosMedData, into experiments as an external test cohort, which contains 1,110 subjects from Russia. This experiment allows us to study the robustness of the system to the factor of race. As shown in Figure 2, the performance (0.9325 AUC) on this dataset is comparable to that (0.9299 AUC) on another external dataset, CC-CCII, whose subjects are all Chinese people. The above experimental results show that the system demonstrates relatively consistent results for different demographic groups, such as age, gender, and race.